# Toxic Effect of Aflatoxins in Dogs Fed Contaminated Commercial Dry Feed: A Review

**DOI:** 10.3390/toxins13010065

**Published:** 2021-01-15

**Authors:** Lizbeth Martínez-Martínez, Arturo G. Valdivia-Flores, Alma Lilian Guerrero-Barrera, Teódulo Quezada-Tristán, Erika Janet Rangel-Muñoz, Raúl Ortiz-Martínez

**Affiliations:** 1Centro de Ciencias Agropecuarias, Universidad Autonoma de Aguascalientes, Aguascalientes 20131, Mexico; lizbeth.martinez@edu.uaa.mx (L.M.-M.); teodulo.quezada@edu.uaa.mx (T.Q.-T.); janet.rangel@edu.uaa.mx (E.J.R.-M.); raul.ortiz@edu.uaa.mx (R.O.-M.); 2Centro de Ciencias Básicas, Universidad Autonoma de Aguascalientes, Aguascalientes 20130, Mexico; alguerre@correo.uaa.mx

**Keywords:** *Aspergillus flavus*, aflatoxicosis, biomarkers, blood coagulation, hepatic diseases

## Abstract

Since its first patent (1897), commercial dry feed (CDF) for dogs has diversified its formulation to meet the nutritional needs of different breeds, age, or special conditions and establish a foundation for integration of these pets into urban lifestyles. The risk of aflatoxicosis in dogs has increased because the ingredients used to formulate CDF have also proliferated, making it difficult to ensure the quality required of each to achieve the safety of the entire CDF. This review contains a description of the fungi and aflatoxins detected in CDF and the ingredients commonly used for their formulation. The mechanisms of action and pathogenic effects of aflatoxins are outlined; as well as the clinical findings, and macroscopic and microscopic lesions found in aflatoxicosis in dogs. In addition, alternatives for diagnosis, treatment, and control of aflatoxins (AF) in CDF are analyzed, such as biomarkers of effect, improvement of blood coagulation, rate of elimination of AF, control of secondary infection, protection of gastric mucosa, reduction of oxidative stress, use of chemo-protectors, sequestrants, grain-free CDF, biocontrol, and maximum permitted limits, are also included.

## 1. Introduction

Commercial dry feed (CDF) for dogs was first patented in 1860 by James Spratt, but it was not until 1957 that it started to be sold commercially. The diversity in the CDF formula increases due to the rise in the knowledge of the nutritional needs of dogs depending on their breeds, ages, and activities carried out. Another aspect that drives the supply of CDF is the increasing availability of agro-industrial ingredients of diverse bromatological composition, which guides the diversification of suitable feed formulas for dogs with differences in their digestive needs, nutrient profiles, and metabolism [1]. Another aspect that drives the demand for CDF is the integration of the dog into the urban lifestyle, as well as the strengthening of the human–pet bond in which dog owners give equal priority to the healthy diet of their pets as well as their own [2].

The CDF offer includes dry derivatives of all types of meat and guts from chicken, beef, pork, and other species; quail, pheasant, and ostrich meat are even incorporated in the making of these feeds [3]. CDF also adds various cereals, such as corn, rice, wheat, barley, and sorghum due to their low cost and acceptable nutritional value; besides this, it does not affect the palatability and digestibility of the nutrients, although the quality and safety of cereals is sometimes objectionable [4]. These ingredients are widely used as a source of energy, as well as a supply of some vitamins, minerals, fibers, and fats [5]. In addition, protein ingredients from soybeans and soybean paste are incorporated as well as some high-fiber ingredients such as alfalfa and oatmeal [6].

Products may be contaminated with *Aspergillus* spp. fungus, aflatoxins (AF), or some of their secondary metabolites as well as other mycotoxins; contamination occurs at various stages of ingredient production, such as flowering, harvest, processing, or storage of cereals as well as metabolic residues in meat, dairy, and egg products [7,8,9]. *Aspergillus* species are mainly found in tropical and subtropical regions around the world. However, due to the global import of food materials and climate change, it is likely that soon also regions with temperate climates (such as Europe) will be as affected as developing countries [10].

As presented later in this review, contamination by *Aspergillus* spp. or its metabolites is an important factor because it reduces the nutritional value of the substrate and produces a great variety of toxic effects; consequently, regulations have been established throughout the world on the maximum permissible levels of aflatoxins in food and feed. Reports of outbreaks of clinical forms of aflatoxicosis in dogs are scarce, but their geographic distribution is very diverse: North America, Latin America, Asia, and Africa. This coincides with a worldwide distribution of aflatoxigenic fungi and their aflatoxins in both complete dog feed and ingredients [11]. Therefore, this work aims to review the presence of aflatoxigenic fungi and their metabolites in commercial dry feed for dogs. The mechanisms of action of AF and its repercussions on animal health are examined. Viable alternatives for the diagnosis and therapy of aflatoxicosis are analyzed; in addition, the control of CDF contamination by aflatoxins is highlighted.

## 2. Methods

This review was carried out by establishing a research objective, search strategies and relevant research articles; a selection of articles, data extraction, data mapping and summary of results were carried out [12]. The literature for this review was identified by searching online databases (Google scholar, PubMed, and Web of Science). Scientific publications were searched from 2000 to 2020. The search terms were ‘aflatoxin’, ‘dog’, ‘food or feed’, and ‘flavus’. All relevant scientific publications were included in the review, but other kind of information was excluded from the analysis. Two researchers independently searched the literature. Then, the two sets of literature were compared; disagreements about the inclusion of the literature were resolved through group discussion to make the decision. Data on the design, objectives, sample size, setting, instrumental methodology, and main findings were extracted. The articles were classified in the following research areas: ‘*Aspergillus* spp.’, ‘Aflatoxins’, ‘Feed ingredients’, ‘Commercial dry feed’, ‘Clinical findings’, ‘Lesions’, ‘Therapeutic strategies’ and ‘Control’. All findings and statements in this review are based on published information as indicated in the references.

## 3. Presence of *Aspergillus* spp. in Commercial Dry Feed for Dogs

The presence of toxigenic *Aspergillus* spp. in cereals or other dog feed ingredients incorporated into CDF formulations deteriorates the quality and affects the safety of the whole feed [13]. Cereals and other ingredients are suitable substrates for the growth of fungal microflora, both in the pre-harvest stage and in storage [14]. When the raw extruded material for CDF formulation presents an excess of relative humidity content (20–25%) in the initial stage of the process, and it is reduced by drying (8–10%), only the growth of the vegetative forms of the fungal microflora is inhibited, but the spores and mycotoxins produced within the processed material remain stable. It has been reported that the optimal temperature and water activity for regulatory genes of the AF biosynthesis pathway to reach their maximum expression are 28–30 °C and 0.96–0.99, respectively, although the AF production range is very wide [15,16,17]. Therefore, if the substrate is rehydrated during improper storage conditions, AF concentrations may increase.

Some studies report that the most frequent fungal contaminants present in CDF for dogs are *Aspergillus* spp., *Mucor* spp., *Penicillium* spp., and *Rhizopus* spp. Several of the genera and species isolated and identified are mycotoxigenic, which produces a risk to dogs’ health [11,18,19,20] (Table 1). Most of the samples included in each study were contaminated by *Aspergillus* spp. and by some other type of toxigenic fungal microbiota. Although information regarding the presence of *Aspergillus* spp. and other fungal microflora in various ingredients of human food is extensive, studies of this contamination in CDF are scarce, despite being made with similar ingredients. Furthermore, these reports refer to contamination of whole feed, but no results of toxigenic microflora in CDF ingredients have been reported [18,19,20,21].

## 4. Aflatoxins and Their Biotransformation Products

The AF are difuranocoumarin compounds produced as secondary metabolites of fungi of the genus *Aspergillus* spp. following a polyketide path; *A. flavus* is the main species of fungus involved in AF production [9,11,22]. Four AF naturally present in agricultural products are described (AFB_1_, AFB_2_, AFG_1,_ AFG_2_); other forms of AF are derived from the metabolic process of these primary forms within the human or animal body [28]. AF are not destroyed by boiling nor do they confer, color, aroma, or flavor to contaminated ingredients, so they usually go unnoticed by both the owner and the dog [13]. When dogs ingest CDF made with ingredients contaminated with AF, the mycotoxin are absorbed in the duodenum and bind to plasma albumin and proceed to be transported through the bloodstream [29].

In various tissues, especially in the liver and kidneys, AF are biotransformed and bioactivated by isoenzymes of the multiple function oxidase system or cytochromes (CyP_450_), giving rise to highly active electrophilic forms called 8,9 endo-epoxide and 8, 9 exo-epoxide, which bind and exert an electrophilic attack on subcellular structures [30]. When AF are metabolized (hydrolysis, demethylation, or ketoreduction), they form less toxic intermediate compounds with greater solubility in water (AFM_1_, AFM_2_, AFQ_1_, AFQ_2,_ AFP_1_, AFP_2,_ and aflatoxicol); therefore, they are eliminated through feces, urine, milk, or egg [31,32]. The most common ways of elimination of metabolites in the urine of dogs is AFM_1_, as well as traces of AFQ_1_ [33,34]. When the epoxide binds to DNA, then an AF-DNA adduct (dihydro-8-(N^7^-guanyl)-9hydroxy-1 AF-N^7^–guanidine) is formed, which rearranges as AF-formaminopyrimidine (AF-FAPy), or it is excreted through urine as AF-N^7^guanine, which is considered a biomarker of genotoxic damage from AF [31,32,33,34]. In general, all compounds derived from natural forms of AF, due to the activity of the enzymes that participate in the detoxification process, are considered biomarkers of exposure and damage [35].

An important detoxification mechanism in many animal species is the involvement of a group of enzymes called Glutathione S-transferases (GST) [36]. The function of GST is to bind the epoxide with the reduced glutathione tripeptide (GSH), which loses two amino acids (glycine and glutamate) to be eliminated as a cysteine residue linked to AF, called mercapturic acid or N-acetylcysteine-AF, which is eliminated through bile or urine [37]. Dogs have a reduced GST activity, which makes them especially susceptible to AF damage; in addition, a deficiency of GSH or its three precursor amino acids facilitates the occurrence of the most extensive oxidative injury [38,39,40].

## 5. Contamination of Feed Ingredients by Aflatoxins

Cereals are usually integrated into dog feeds, especially corn, sorghum, rice, wheat, oats, barley, and millet; they are a good source of carbohydrates, fiber, protein, fat, minerals, and vitamins [41,42]. However, cereals present an important risk for the health of dogs because they are vulnerable to contamination by *A. flavus* both in the field and in storage [43,44,45]. In some CDF formulations for dogs, pumpkin seeds, chia, quinoa, and even some legumes such as lentils are included among the ingredients due to their high protein and mineral content. However, these ingredients can also be contaminated by some forms of AF [46,47,48]. Furthermore, the incorporation of both potatoes and sweet potatoes in the manufacturing of premium types of dog feeds is used as a source of carbohydrates and fiber. It is reported that the presence of *A. flavus* and high concentrations of AF can be found in potato tubers [49,50].

Fruits in CDF are used as a natural source of fiber; papaya is one of the main fruits that are included, however, it may have *A. flavus* in the postharvest, which has effects on its nutritional value, and it may also have AF concentrations [51]. Blueberries are used as antioxidants, although there are reports of *A. flavus* contamination and concentrations of AFB_1_, AFB_2_, y AFG_1_ [52]. Orange and coconut are other ingredients that are included in some dog feeds, but they can also be contaminated by FA-producing fungal microflora [53,54,55].

Dairy and meat products as well as eggs are added to dog feed as an important source of proteins and fats. However, secondary AF metabolites such as AFM_1_, AFM_2_, AFP_1_ may be found, which can also contaminate these feed ingredients [56]. The AF residues can be located in by-products of animal origin used in the manufacturing of CDF, such as liver, kidneys, muscle, meat, milk, and egg. The residual compounds in eggs, milk and meat are derived from the biotransformation of the original AF ingested in the feed of animals used for food and remain in the dog that ingests the CDF [57].

## 6. Aflatoxin Contamination in Commercial Dry Feed for Dogs

Natural forms of AFs and its metabolites can be found in the ingredients used to make CDF for dogs (Table 2). Cereals used in the formulation of CDF may contain high levels of AF contamination (0.48–1.081 μg/kg), making them the most likely sources of aflatoxin contamination [58,59]. AFB_1_ is the most abundant form in open sampling CDF trials, with values that vary widely (<0.5 and 4.946 μg/kg) [8,60]. CDF is classified into economic, premium, or super premium types of dog feeds according to the nutritional quality of the ingredients, but this classification does not guarantee that it is an AF-free product because they are found in all types of CDF [20,61,62]. Therefore, AF present in the CDF are a health risk, which is especially important because CDF is used as the sole or main component of the diet during most of a dog’s life; in addition, all the feed contained on each bag is usually eaten until it is exhausted, suggesting that prolonged ingestion of feed contaminated with these mycotoxins, even at low doses, can have adverse health effects [63].

## 7. Clinical Findings in Dog Aflatoxicosis

The median lethal dose (LD_50_) of AFB_1_ for the dog is of 0.5–1.5 mg/kg body weight; clinical manifestations are even observed at doses greater than 60 µg/kg of AF in feed [73]. As can be seen in Table 3, higher doses are associated with acute forms of aflatoxicosis. In addition, studies of poisoning outbreaks in dogs have found very high AFB_1_ values (<5.0 mg/kg) [66].

The clinical signs shown by dogs with aflatoxicosis are identified as digestive, hemodynamic, and nervous alterations. Digestive findings include vomiting, anorexia, hematemesis, hematochezia, and melena. The hemodynamic changes reported are ascites, peripheral edema, jaundice, dehydration, decreased blood pressure, hemorrhagic diathesis, and petechiae in the mucous membranes [38,75,76,77]. Clinical findings related to central nervous system disturbances are attributable to hepatic encephalopathy, manifested by depression, vocalization, stupor, seizures, and coma [38,77].

The clinical findings observed in cases of aflatoxicosis are also related to some variables of the blood biochemistry and coagulation tests (Table 4); the changes that occur are the increase in the specific activity of the liver enzymes: alanine aminotransferase, aspartate aminotransferase, and alkaline phosphatase (ALT, AST, and ALP). The increased activity of these hepatic membrane enzymes is a biomarker of damage due to injury induced by the epoxides generated via AF metabolism. Coagulation tests in cases of aflatoxicosis show a decrease in the ability of blood to clot, evidenced mainly by an increase in prothrombin time and activated partial thromboplastin time (PT and aPTT) as well as a decrease in antithrombin in plasma, protein C activity, and coagulation factor VII (FVII: C) [78]. A decrease in fibrinogen and platelets is also observed. The hemorrhagic effects of AF are attributed to its chemical structure that contains a coumarin ring with an anticoagulant effect; therefore, a delay in coagulation occurs and secondarily induces disseminated intravascular coagulation (DIC) with depletion of the coagulation factors [38,74].

In aflatoxicosis, hyperbilirubinemia is also seen, which is associated with liver failure or inability to conjugate bilirubin generated in the spleen in higher-than-normal amounts, resulting in an increase in total bilirubin [66]. In acute AF poisonings, a decrease in total proteins and albumin is observed. These changes are related to impaired liver function, as well as protein loss from enteric hemorrhage, ascites, and edema [80]. The decrease in cholesterol concentrations in AF poisonings is due to cholestasis, which arises from fibrosis of the bile ducts [81]. Therefore, the digestive and nervous clinical signs that are present simultaneously with the enzymatic and hematic changes specified, suggest the presence of aflatoxicosis, which must be corroborated with clinical history data and with the presence of AF in the feed and in the dog’s stomach. In general, biochemical analyses and coagulation tests are used for the early detection of the effects of exposure to AF; therefore, they constitute biomarkers of exposure and damage by AF [82].

## 8. Macroscopic and Microscopic Lesions in Aflatoxicosis in Dogs

When dogs ingest CDF contaminated by AF, bioactive compounds are generated that attack subcellular structures and cause damage [30]. For this reason, various macroscopic and microscopic lesions are presented that can be used as a suggestive diagnostic element. The most reported findings are restricted to the liver; however, localized lesions in other organs are also reported, which are related to direct damage from AF, liver failure, coagulation abnormalities, or disseminated intravascular coagulation (Table 5) [66,75,76].

## 9. Therapeutic Strategies

When a diagnosis of aflatoxicosis has been established in dogs, it is suggested that therapy include, in addition to symptomatic treatment, hemostatic stabilization, increasing the rate of AF elimination and hepatic–renal protection (Table 6) [33,38]. Because AF have an anticoagulant effect like coumarin, the provision of vitamin K_1_ is proposed to promote the activation of coagulation factors II, VII, IX, X, protein C, and protein S [73,82,88], as well as the provision of intravenous plasma to provide procoagulant proteins [88]. To increase the rate of elimination of AF metabolites, intravenous fluid therapy is administered [33], which is also indicated to correct dehydration and hypovolemia that occur in these cases. As symptomatic treatment, antiemetics, such as metoclopramide or ondansetron; protectors of the gastroenteric mucosa, such as famotidine and sucralfate; and broad-spectrum antibiotics are administered. The latter are used to protect against systemic infection caused by AF-mediated immunosuppression.

The use of N-acetylcysteine (NAC) in dogs with AF poisoning is successful [91] because this compound provides an essential amino acid (L-cysteine) for the intracellular synthesis of GSH; this tripeptide is bound to reactive AF epoxides by the interaction of the GST enzyme, and the combined AF-GSH compound is eliminated as mercapturic acid in feces and urine [92]. Furthermore, NAC functions by itself as a free radical scavenger and has anti-inflammatory properties [91]. The use of silymarin is also proposed [84]; this flavonoid derived from milk thistle could increase GST activity and promote GSH’s synthesis [88]. When the rate of elimination of reactive AF metabolites is increased, the attack of subcellular structures is reduced or inhibited and therefore, cellular integrity is protected, especially of the liver and kidney as targeted organs of attack by AF [89]. Vitamin E is another liver protector used in CDF to prevent lipid peroxidation by AF epoxides and to prevent damage to cell membranes. In the same sense, the use of L-carnitine reduces oxidative damage; this amino acid transports fatty acids from the cytosol to the mitochondria for their β-oxidation and energy generation, thereby reducing intracellular lipid deposition and protecting the cell membrane against the epoxide-induced lipid peroxidation processes of AF [93].

## 10. Methods Used to Control Aflatoxins in Commercial Dry Feed

The dog feed industry uses various physicochemical and biological methods to reduce AF contamination. Physical processing techniques of CDF ingredients are successfully used, including sieving and pearling of cereals, which are used to separate damaged grains and abrasion of the outer portions of the seeds; both methods decrease the growth of fungal microflora and reduce the content of AF [94]. The use of washing techniques for cereal grains is suggested; although the product moistened by this method requires being completely dried before storage, which generates an additional cost [95]. The proper extrusion of the grains eliminates fungal spores present in the mixture of raw materials because they cannot survive at the temperature and pressure used (150 °C, 37 atm) [96].

They also use fungal microbiota inhibitors based on a wide variety of chemical compounds. Benzoic, acetic, sorbic, and propionic acids inhibit the growth of fungi by acidifying their cytoplasmic content, which is why it is a method used to prevent the formation of AF [94]. The use of ozone gas (O_3_) in DCF is proposed as an oxidizing agent because O_3_ alters the structure of cell membranes and induces alterations in cell permeability and destruction of the fungal microflora. Furthermore, when O_3_ is applied to feed, a reaction occurs with the double bond C8 = C9 to form a vinyl ether in the furan terminal ring of AF; then, an intermediate compound called AF-ozonide is formed; finally, the AF-ozonide is easily degraded into less toxic compounds (carboxylic acid, aldehyde, ketone, and carbon dioxide) [96,97,98,99].

The bioavailability of AF can be decreased by using compounds that reduce its gastrointestinal absorption [31]. These sequestering compounds are widely used because they bind to AF within the digestive tract of dogs through the chemisorption of β-dicarbonyl from AF, thereby reducing their intestinal absorption. The most widely used mineral sequestrants are some phyllosilicates, such as hydrated calcium and sodium aluminosilicates, bentonite, and tectosilicates or zeolites [100]. Bentonite is administered to absorb AF through cation exchange and carbonyl groups, as well as ion-dipole interaction [100]. The adsorption of the zeolite arises from the interaction of the AF with Ca^2+^ existing on its surface [101]. Furthermore, the use of organic compounds derived from yeast and other microorganisms is a method of adsorption of AFs present in CDF for dogs. *Saccharomyces cerevisiae* cell walls contain large proportions of mannoproteins and β-D-glucomannan, which exhibit high activity to adsorb AF. Lactic acid bacteria also could reduce the bioavailability of AF through their adsorption by peptidoglycans from their cell wall [102].

The methods of reducing the bioavailability of AF in feed have a wide range of efficacy, but they do not eliminate the risk of poisoning at high concentrations of AF, or long-time ingestion, and they do not prevent the development of fungi and an increased AF concentration when CDF is improperly stored [63]. Even AF contamination has not been able to be eliminated in the food supply chain, even if good agricultural practices are adhered to or optimal storage conditions are maintained. Although the best conditions of agro-industrial production of CDF are able to ensure that the concentration of AF is innocuous and it is distributed within airtight sacks, as soon as the sack is exposed to the environment, the hygroscopicity increases the relative humidity of the CDF (>17%) and the conidia can germinate and begin the production of AF to toxic levels. This risk may be greater with increasing ambient temperature, relative humidity, and time of consumption of the sack content [16,17,103].

One of the most common strategies to control AF contamination is to set the maximum residue level (MRL) or the action levels for AF, which are the maximum concentrations permitted of AF in food or feed [104]. The MRL for AF in feedstuffs varies widely among different countries (0.0–50.0 µg/kg) and is indicated for any animal feed. The European Union has fixed the MRL for AFB_1_ at 10.0 μg/kg in complete feed and twice as much for feed materials; China, Japan, and Korea apply the same MRL (10.0 μg/kg) for animal feed, especially in dairy feed. However, in many countries in both North America and Latin America, the MRL for AF (aflatoxin B_1_ + B_2_ + G_1_ and G_2_) are set at 20 µg/kg for AF in feedstuffs because it is assumed that animals like dogs are susceptible to the toxic effects of AF at higher doses [73,105]; although, many countries of Asia and Oceania, like India, Nepal and Senegal have higher MRL (>20.0 μg/kg) for AF in feedstuffs, including in the dog feed [63,95,104,106]. On other hand, these regulations are intended to protect animal health and prevent AF toxicity; but there is no evidence of the effect that prolonged exposure to AF below the MRL could have on dog health, especially when AF-contaminated CDF is ingested until all the feed contained within each bag is finished, suggesting an urgent need for quality control mechanisms.

## 11. Conclusions and Recommendations

Aflatoxins and the fungi that produce them, particularly *Aspergillus flavus*, are common in the main ingredients used to make commercial dry feed for dogs (grains, meat and bone meals, viscera, tubers, fruits, etc.). These ingredients are agro-industrial by-products that are used to satisfy the specific nutritional requirements of the dog (age, weight, activity, etc.) and to allow its incorporation into the urban lifestyle. Therefore, the presence of mycotoxigenic fungi and their toxins could be considered a serious problem for the health of the dog, to develop its zootechnical function as a companion, guardian, or as a sports animal. In addition, dangerous levels of aflatoxins are potentiated because the dog must ingest large amounts of the CDF until all the content found in each bag that its owner purchases is finished off. Although studies on the impact of exposure to AF on the health of dogs are limited, these studies are adequate to demonstrate its impact on clinical manifestations, macroscopic and microscopic lesions, and hematological and enzymatic alterations. Therefore, to address these problems, the use of therapeutic and control strategies that mitigate the impact developed by aflatoxins is recommended. In addition, the establishment of maximum permissible levels of AF specifically for CDF and research on prolonged exposure to low concentrations of aflatoxins should be encouraged.

## Figures and Tables

**Table 1 toxins-13-00065-t001:** Fungal microflora detected in commercial dry feed for dogs.

Location	Number of Samples (*n*)	Major Fungi Identified	Positive Samples (%)	Citation
Argentina	12	*Aspergillus flavus*, *A. niger*, *Mucor globosus*, *M. plumbeus*, *M. racemosus*, *Rhizopus* spp.	100	[11]
Brazil	180	*A. flavus*, *A. candidus*, *A. flavipes*, *A. fumigatus*, *A. niger*, *A. ochraceus*, *A. parasiticus*, *Cladosporium* spp., *Fusarium* spp., *Mucor* spp., *Penicillium* spp.	100	[22]
Brazil	34	*Absidia* spp., *Aureobasidium* spp., *Alternaria* spp., *Aspergillus* spp., *Chrysonilia* spp., *Cladosporium* spp., *Emericella* spp., *Eurotium* spp., *Fusarium* spp., *Geotrichum* spp., *Monascus* spp., *Mucor* spp., *Olyptrichum* spp., *Paecilomyces* spp., *Penicillium* spp., *Phoma* spp., *Rhodotorula* spp., *Rhizopus* spp., *Scapulariopsis* spp., *Syncephalostrum* spp., *Tilletiopsis* spp., *Trichoderma* spp., *Wallemia* spp., *Yeasts*	74	[23]
Brazil	60	*Aspergillus* spp., *Fusarium* spp., *Penicillium* spp., *Rhizopus* spp.	53.3	[24]
Poland	25	*Aspergillus* spp., *Mucor* spp., *Penicillium* spp.	52	[19]
Poland	20	*Aspergillus* spp., *Penicillium* spp., *Rhizopus* spp.	25	[21]
Poland	25	*Aspergillus* spp., *Alternaria* spp., *Cladosporium* spp., *Fusarium* spp., *F. verticillioides.*, *F. proliferatum*	100	[18]
Portugal	20	*A. niger*, *Mucor* spp., *Penicillium* spp.	100	[25]
South Africa	20	*A. flavus*, *A. fumigatus*, *A. parasiticus*, *F. graminearum*, *F. verticilloides*, *Penicillium* spp.	100	[20]
United Kingdom	5	*Absidia* spp., *Acremonium* spp., *Alternaria* spp., *Aspergillus* spp., *Cladosporium* spp., *Eurotium* spp., *Mucor* spp., *Pénicillium* spp., *Rhizopus* spp., *Syncephalastrum* spp., *Wallemia* spp., *yeasts*.	100	[26]
Venezuela	4	*Acremonium charticola*, *A. flavus*, *A. fumigatus*, *A. terreus*, *C. herbarum*, *F. poae*, *P. citrinum*, *P. expansum.*	94	[27]

**Table 2 toxins-13-00065-t002:** Presence of aflatoxins in commercial dry feed for dogs.

Location	Number of Sample (*n*)	Test	Mean AF (μg/kg)	Positive Samples (%)	Citation
Brazil	45	TLC	AFB_1_ (19.0)	AFB_1_ (6.7)	[64]
Brazil	180	HPLC	AFB_1_ (7.0)	AFB_1_ (100)	[22]
Brazil	(AE) 49 (AP) 25 (ASP) 13	HPLC	(SF) AF (1.2) (PF) AF (0.4) (SPF) AF (0.5)	AF (95.4)	[61]
China	32	LC-MS/MS	AFB1 (47.7)	AFB_1_ (87.5)	[65]
United States	9	ELISA, TLC, HPLC	AFB_1_ (530) AFB_2_ (19.0)	AFB_1_ (88.8) AFB_2_ (77.7)	[66]
Italy	(AE) 24 (AP) 24	LC-MS, PLC-MS/MS	AFB_1_ y AFG_1_ (<0.5) AFB_2_ (5.7) AFG_2_ (15.8)	AF (12.0)	[62]
Italy	55	UHPLC-Q-Orbitrap HRMS	AFB_1_ (4.3)	AFB_1_ (25.8)	[67]
Mexico	19	HPLC	AFB_1_ (5.0), AFB_2_(0.07), AFG_1_ (0.05), AFG_2_ (0.03), AFM_1_ (2.0) AFM_2_ (0.1) AFP_1_ (1.1), AFL (0.3)	AFB_1_ (79.0), AFB_2_ (26.0), AFG_1_ (63.0), AFG_2_ (21.0), AFM_1_ (63.0), AFM_2_ (89.0), AFP_1_ (58.0), AFL (47.0)	[68]
Mexico	29	HPLC-FL	AFB_1_ (1.6), AFB_2_(0.1), AFG_1_ (28.2), AFG_2_ (1.3), AFM_1_ (1.8), AFM_2_ (0.2), AFP_1_ (1.7), AFL (28.6)	AFB_1_ (76.0), AFB_2_ (4.0), AFG_1_ (86.0), AFG_2_ (93.0), AFM_1_ (48.0), AFM_2_ (21.0), AFP_1_ (100), AFL (100)	[8]
Nigeria	30	HPLC	AF (9.6)	AF (100)	[69]
Poland	25	HPLC-FLD	AF (0.2)	AF (4.0)	[19]
South Africa	(AE)10 (AP)10	TLC, HPLC-FLD	(SF) AFB_1_ (44.1) (PF) AFB_1_ (20.1)	AFB_1_ (100)	[20]
Turkey	21	ELISA	AFB_1_ (6.6)	AFB_1_ (100)	[70]
Turkey	18	ELISA	AF 1.75 a 20	AF (16.7)	[71]
Brazil	Retrospective study	HPLC	AFB_1_–AFG_1_ (89.0–191)	-	[72]

Definitions: AF: Total aflatoxins; AFB_1_: Aflatoxin B_1_; AFB2: Aflatoxin B_2_; AFG_1_: Aflatoxin G_1_; AFG_2_: Aflatoxin G_2_; AFL: Aflatoxicol; AFM_1_: Aflatoxin M_1_; AFM_2_: Aflatoxin M_2_; AFP_1_: Aflatoxin P_1_; ELISA: Enzyme Linked Immunosorbent Assay; HPLC: High Performance Liquid Chromatography; LC-MS/MS: Liquid Chromatography-Tandem Mass Spectrometry; LC-MS: Liquid chromatography coupled to mass spectrometry; PLC-MS/MS: Ultra performance liquid chromatography coupled to tandem mass spectrometry; UHPLC-Q-Orbitrap HRMS: ultra-high performance liquid chromatography coupled to high resolution mass spectrometry; HPLC-FL: Fluorescence High Performance Liquid Chromatography; HPLC-FLD with fluorescence detection (FLD); SF: standard feed; PF: premium feed; SPF: super premium feed.

**Table 3 toxins-13-00065-t003:** Acute poisoning in dogs due to ingestion of aflatoxins in commercial dry feed (CDF).

Location	Number of Intoxicated Dogs (*n*)	Mortality (%)	Range of AF in CDF (µg/kg)	Citation
Brazil	4	100	AFB_1_–AFB_2_ (89.0–191)	[72]
Brazil	2	100	AFB_1_ (83.2–150)	[74]
Israel	50	68.0	AF (80–300)	[38]
South Africa	10	100	AF (100–300)	[75]
South Africa	100	96.0	AF (<5–4946)	[76]
United States	9	100	AFB_1_ (223–579)	[66]
United States	72	36.1	AF (48–800)	[77]

**Table 4 toxins-13-00065-t004:** Blood biochemistry and coagulation tests in aflatoxicosis in dogs *.

Analyte	A. Intoxication	B. Reference Values	Comparison (A/B)
**Blood biochemistry**			
Total bilirubin (µM/L)	130 (1.71–428)	2.7 (0.00–5.1)	48.1 (0.00–83.9)
ALT (U/L)	598 (6.0–2278)	59.5 (5.0–106)	10.1 (1.20–21.5)
AST (U/L)	178 (15.0–748)	31.8 (9.0–56.0)	5.6 (1.7–13.4)
ALP (U/L)	284 (10.0–3477)	67.3 (4.0–140)	4.2 (2.5–24.8)
TP (s)	41.4 (4.5–71)	12.0 (6.0–18.0)	3.4 (0.75–3.9)
APTT (s)	34.5 (9.6–241)	15.7 (10.0–23.8)	2.2 (0.96–10.1)
GGT (U/L)	10.4 (0.00–44.4)	7.8 (0.00–19.0)	1.34 (0.00–2.3)
Total protein (g/dL)	4.9 (1.10–7.9)	6.3 (5.4–7.1)	0.78 (0.20–1.11)
Albumin (g/dL)	2.5 (0.50–3.9)	3.5 (2.8–4.1)	0.70 (0.18–0.95)
**Coagulation tests**			
FVII:C (% activity)	32.0 (1.40–67.0)	125 (50–200)	0.26 (0.03–0.34)
Platelets (X10^9^/L)	156 (8.0–432)	347 (143–700)	0.45 (0.06–0.62)
Protein C (% activity)	18.0 (4.0–55.0)	105 (75–135)	0.17 (0.05–0.41)
Cholesterol (mmol/L)	1.26 (0.00–7.9)	5.2 (2.6–8.6)	0.24 (0.00–0.91)
Antithrombin (% activity)	11.0 (0.00–147)	105 (65–145)	0.10 (0.00–1.01)
Fibrinogen (mg/dL)	37.0 (11.0–344)	305 (100–510)	0.12 (0.11–0.67)

Definitions: ALT: Alanine aminotransferase; AST: Aspartate aminotransferase; ALP: alkaline phosphatase; PT: prothrombin time; APTT: active partial thromboplastin time; GGT: γ-glutamyltransferase; FVII: C: Coagulation factor VII. * Mean and range values adapted from: [38,66,77,79].

**Table 5 toxins-13-00065-t005:** Macroscopic and microscopic lesions in aflatoxin poisoning in dogs *.

Location	Macroscopic Lesions	Microscopic Lesions
Generalized	Hemorrhagic diathesis, jaundice, and ascites.	
Hepatic	Hepatomegaly, uneven surface, pale yellowish discoloration, enhanced lobular pattern, cholestasis, and gallbladder edema.	Hepatocytes with micro and macrovesicular steatosis, cytomegaly, pyknosis, karyorexis, and necrosis. Centrilobular areas with hemorrhage, reticulin, and collagen. Hyperplasia and proliferation of bile ducts.
Pulmonary	Atelectasis, congestion, pleural effusion, hydrothorax, and petechiae.	Alveoli with hemorrhage and perivascular edema.
Cardiac	Ecchymosis and petechiae in the endocardium and epicardium. Hydropericardium.	
Digestive	Edema and congestion in the gastrointestinal lumen, mesenteric lymph nodes and pancreas.	Necrosis with mononuclear infiltration in the mucosa.
Splenic	Splenomegaly	Diffuse perivascular edema and red pulp with hemorrhages and erythrophagocytosis.
Renal	Dark red coloration with subcapsular depressions and multifocal hemorrhages.	Fluid accumulation in Bowman’s space and glomerular basement membrane thickening. Multifocal vascular congestion in the interstitial tissue of the renal medulla and pelvis. Degeneration, ectasia, and necrosis of the proximal and distal tubular epithelium.

* Adapted from [66,74,75,76,83,84,85,86,87].

**Table 6 toxins-13-00065-t006:** Suggested therapy for aflatoxicosis treatment in dogs *.

Drug	Dose	Administration via	Usual Interval (h/d)	Therapeutic Indications
**Hemostatic Stabilization**				
Vitamin K_1_	2.0 mg/kg	SC	24/5	Synthesis of coagulation factors
Intravenous plasma	10.0 mL/kg	IV	Until TP & TTPa are restored	Correction of coagulopathy
**Elimination of AF and Hepato-Renal Protection**				
Hartman solution	40.0–60.0 mL/kg/d	IV	cbp	Increased glomerular filtration rate and restoration of water/electrolyte balance
N-acetylcysteine	70.0 mg/kg	IV	12/15	GSH synthesis and AF binding
Silymarin	20 mg/kg	PO	24/30	Counter AF epoxides
Vitamin E	10.0 U/kg	PO	24/30	Counter AF epoxides
L-carnitine	50.0–100 mg/kg	PO	8/30	Decreased liver lipidosis and counter epoxides
**Symptomatic Treatment**				
Metoclopramide	0.40 mg/kg	SC	8/3	Antiemetic
Ondansetron	0.15 mg/kg	IV	12/3	Antiemetic
Famotidine	0.5 mg/kg	IV	12/30	Decreased gastric secretion
Sucralfate	0.5–1.0 g/dog	PO	8/30	Gastric cytoprotection
Ampicillin	25.0 mg/kg	IV	8/7	Broad spectrum bactericide
Enrofloxacin	5.0–20.0 mg/kg	IV	12/7 días	Broad spectrum bactericide

Definitions: TP: prothrombin time; TTPa: activated partial thromboplastin time. * Adapted from [38,77,88,89,90].

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
