# Peer review of "Toxic Effect of Aflatoxins in Dogs Fed Contaminated Commercial Dry Feed: A Review"

_toxins, 2021, doi:10.3390/toxins13010065_

Round 1
Reviewer 1 Report
This is an interesting and well referenced review of the prevalence and toxicology of aflatoxins in dog food and its supply chain ingredients. The clinical features and treatment of aflatoxicosis in dogs are well addressed and reduction measures for aflatoxins in feed are touched on. The review is an accurate representation of the cited papers and presents a coherent overview of this important topic.
The introduction section of the papers is missing and the cited papers begin at reference 11.
There is no description of the literature search strategy nor of the inclusion and exclusion criteria for the papers cited and if this is to hand it should be briefly included.
In section 6 page 4 (of 12) line 131 it is stated “ …, no maximum limits have been established for the permissible content 131 of AFs in dog food [65].” Please see Directive 2002/32/EC of the European Parliament and of the Council of 7 May 2002 on undesirable substances in animal feed. In this Directive (see https://eur-lex.europa.eu/legal-content/EN/TXT/?uri=CELEX%3A02002L0032-20191128&qid=1608718495682 ) feed for “animals belonging to species normally fed and kept or consumed by man …” are regulated, which I take to include commercial dry food (CDF) for dogs. Annex I, Section II, Mycotoxins limits the concentration of aflatoxin B1 in feed materials to 0.02 mg/kg (20 μg/kg) and in complementary and comlete feed to 0.01 mg/kg (10 μg/kg). The authors should reference these limits.
Author Response
Observation |
Location (Line No.) |
Attention to observations |
The introduction section of the papers is missing and the cited papers begin at reference 11. |
|
We do not know why the introductory section of the revised document was removed. The introduction to the manuscript was included as well as the corresponding citations. |
There is no description of the literature search strategy nor of the inclusion and exclusion criteria for the papers cited and if this is to hand it should be briefly included. |
|
Bibliographic search strategies and criteria for inclusion and exclusion of cited articles were included in the revised manuscript |
Please see Directive 2002/32/EC of the European Parliament and of the Council of 7 May 2002 on undesirable substances in animal feed. In this Directive (see https://eur-lex.europa.eu/legal-content/EN/TXT/?uri=CELEX%3A02002L0032-20191128&qid=1608718495682 ) feed for “animals belonging to species normally fed and kept or consumed by man …” are regulated, which I take to include commercial dry food (CDF) for dogs. The authors should reference these limits. |
131 |
The revised manuscript included Directive 2002/32/EC as well as the international regulations on maximum residue levels for aflatoxins that are applicable to dry commercial dog feed. |
Reviewer 2 Report
Though this review covers an interesting and important topic, significant re-organization and greater discussion and synthesis are needed. In addition, the scope of the review should be clearly defined. The information is presented as applying to the commercial dog food industry in general, but the dog food industry and associated practices and regulations must vary considerably among different countries. It is also not clear the extent to which aflatoxin contamination of commercial dog food is a problem or if it varies by country/region. The review should begin by defining which countries/regions the information in the review applies to, establishing the extent of the problem (when and where dog food contamination events have occurred and their severity), and any regulations or feed safety practices that currently exist. The review should conclude with specific suggestions for mitigating the problem both from a research and regulatory perspective. Specific suggestions are included as annotations in the attached PDF.

Author Response
Observation |
Location (Line No.) |
Attention to observations |
New Location (Line No.) |
Significant re-organization and greater discussion and synthesis are needed. In addition, the scope of the review should be clearly defined. |
|
The introductory section of the document was revised, and the entire manuscript was improved with attention to all the observations that were considered important by the reviewers |
|
It is also not clear the extent to which aflatoxin contamination of commercial dog food is a problem or if it varies by country/region. |
|
The revised manuscript indicates that aflatoxin contamination of commercial dog feeds is a problem in tropical and subtropical regions, especially in developing countries, but regions with temperate climates (such as Europe) may become more problematic due to global importation of feed ingredients or as an effect of climate change. |
49-52 297-310 |
The review should begin by defining which countries/regions the information in the review applies to, establishing the extent of the problem (when and where dog food contamination events have occurred and their severity), and any regulations or feed safety practices that currently exist. |
|
The introductory section of the revised manuscript was included a definition of the regions where the information from the review applies, establishing the scope of the problem. Although information on clinical disease outbreaks is scarce, the worldwide presence of the toxigenic fungus and aflatoxins allows us to assume that subclinical forms of aflatoxicosis in dogs have a wide occurrence. |
49-52 Tables 1-3 297-310 |
The review should conclude with specific suggestions for mitigating the problem both from a research and regulatory perspective. |
|
The revised document was concluded with concrete suggestions for mitigating the problem from both a research and regulatory issues. To address these problems, research on therapeutic and control strategies that mitigate the impact developed by aflatoxins is recommended, as well as research on the effects of long-term exposure to low concentrations of aflatoxins. Also, it is suggested to promote the establishment of maximum permitted levels of AF specifically for CDF. |
311-326 |
|
|
|
|
Specific suggestions included in the PDF |
|||
1.- Attach “:” |
3 |
The “:” was attached. |
3 |
2.- I am not sure what you mean here. How has this led to an increase in risk?
|
7 |
The revised manuscript adds that the risk of aflatoxicosis in dogs has increased because the ingredients used to formulate CDF have also proliferated, making it difficult to ensure the quality required of each to achieve the safety of the entire CDF. |
7-8 |
3.- Delete text “etc” |
14 |
The textual expression “etc” was deleted |
15 |
3.- Remove “The review study” |
16 |
Is added “This review” |
18 |
4.- spp. (not italicized - correct throughout) |
28 |
The not-italicized word “spp.” was corrected throughout the document. |
|
5.- This is the first time CDF is used in the main text, so it should be defined first. |
29 |
In the introductory section of the revised document was incorporated the definition of commercial dry feed as CDF. |
29 |
6.- Remove “Contamination by” |
28 |
In the revised manuscript, the expression "Contamination by" was removed and "Presence of" was added. |
79 |
7.- Add “toxigenic” |
28 |
Is added the word “toxigenic” |
79 |
8.- Remove “toxigenic agents” |
28 |
The term "toxigenic agents" was removed from the revised manuscript. |
|
9.- What ingredients? It has not clearly been stated what type of ingredients you are referring to. |
30 |
The ingredients that can be easy contaminated by aflatoxins are described |
80-81 |
10.- Add “affects” |
29 |
The term "affects" was rewritten in the revised manuscript. |
80 |
11.- Do you mean humidity or moisture content?
|
32 |
The term "humidity" was replaced by "relative humidity content" in the revised manuscript. |
83 |
12.- Perhaps you should mention here that it the substrate becomes re-hydrated, mycotoxin concentrations can increase. |
35 |
The revised manuscript explains why mycotoxin concentrations may increase when the substrate is rehydrated. |
85-89 |
13.- Remove “matters” |
37 |
The term "matters" was replaced by "fungal contaminants" in the revised manuscript. |
90 |
14.- Remove “genus” |
39 |
The term "genus" was replaced by "genera" in the revised manuscript. |
91 |
15.- Remove “are” |
|
The term "are" was replaced by "were" in the revised manuscript. |
93 |
16.- This sentence needs to be expanded upon or left out. I am not sure what point is being made here. Is this suggesting these were are low quality products with poor quality control? What are there so few reports of fungi associated with commercial dog food in the literature? Does the lack of reports suggest this is not a widespread problem or just a lack of awareness and regulation? Why are so few countries represented in Table 1? These questions should be addressed somewhere in the review and some sort of overall assessment of the extent of the problem needs to be discussed. |
41 |
The revised document discusses the lack of information on the presence of Aspergillus spp. and other fungal microflora in both the whole feed and various ingredients of CDF. Mention of the association with other bacterial pathogens and chemical contaminants was removed as suggested by the reviewer because it is not the topic to be developed in this manuscript. More reports of fungal microflora detected in commercial dry dog feeds were included |
94-98 |
17.- Remove “authors” |
|
The term "authors" was replaced by "Citation" in all tables of the revised manuscript. |
|
18.- I do not know what the point of this statement is.
|
85 |
The paragraph was reformulated to explain more clearly the reasons for the integration of cereals into dog feed, as well as the high probability of aflatoxin contamination. |
131-134 |
19.- This sentence does not seem necessary. You have already stated that fungal and mycotoxin contamination are a problem in dog food ingredients. However, somewhere the specific concentrations at which aflatoxins cause poisoning or other health effects should be stated. |
86 |
This sentence was reformulated to explain that cereals are the most likely source of AF contamination. This sentence also was relocated in the section about the aflatoxin contamination in CDF (Section 6). This sentence is a basis for addressing in Section 7 of the manuscript (aflatoxicosis in dogs) the specific concentrations at which aflatoxins cause poisoning or other health effects. |
131-134 155-156
|
20.- One of these references is for sweet potatoes. It is not clear here if you are talking about potatoes, sweet potatoes, or both. Also, both of these are unlikely to be a major source of aflatoxin in dog food. In this entire section it is unclear if the various ingredients listed have been confirmed as sources of aflatoxins in dog food or if there are just miscellaneous reports in the literature of the ingredients/crops being contaminated with aflatoxins. What are the most likely sources of aflatoxin contamination in commercial dog food?
|
92 |
It is specified in the document the inclusion of both types of potatoes as an alternative to the use of cereals, precisely because of the susceptibility of the latter to mycotoxin contamination. However, it is clarified that like any ingredient, they could also be contaminated by aflatoxins. |
137-138
|
21.- Is there any evidence that meat and eggs are a major source of aflatoxin contamination in dog food? It seems unlikely. This paragraph makes it seem like any and all ingredients are potentially contaminated with aflatoxins. While this may be technically true, most of these ingredients are extremely unlikely to be a major source of aflatoxins in dog food.
|
102 |
The revised document discusses that dairy and meat products, as well as eggs, are added to the diet of dogs as an important source of protein and fat. However, secondary metabolites of AF such as AFM1, AFM2, AFP1 can be found, which can also contaminate these CDF ingredients because of AF contamination in the food chain. |
150-152 |
22.- Is this statement true everywhere? In the U.S., I assume dog food would fall under the category of "all other" under the FDA's action levels for aflatoxin. I assume that some countries have regulations that apply to dog food even if it is not consistently enforced.
|
131 |
This sentence was also relocated to the section on the regulation of aflatoxin contamination by monitoring of maximum residues allowed in animal feed. This sentence was reformulated to explain the regional variation of aflatoxin regulations in animal feed. |
302-305 |
23.- As alluded to in other sections, I think you should make clear the scope of your review. What countries/regions are you referring to? The dog food industry and associated practices and regulations must vary considerably among different countries.
|
216 |
International regulations were analyzed for maximum residue levels (MRL) or action levels for AF, which are the maximum allowable concentrations of AF in food or feed. |
49-52 297-310 |
24.- Is this correct? I looked up the references you cite here, and I do not see any information regarding detoxification of aflatoxin with ozone.
|
231 |
The revised manuscript describes the degradation of AF in CDF by O3 to less toxic compounds, which reduces the risk of AF toxicity. New references to the topic are added. |
270-274 |
25.- A brief discussion of storage conditions that would lead to increases in aflatoxin should be provided. How likely is this to occur prior to and/or after purchase of the dog food by the consumer? |
248 |
Analysis of storage conditions that would lead to increased aflatoxin production is included. |
291-296 |
Reviewer 3 Report
Authors reviewed the issue of aflatoxins contamination in dog food, providing an overview of the mycotoxigenic fungal species and aflatoxins occurrence detected in commercial feed and in the ingredients commonly used for its formulation; an additional discussion on the mechanisms of action and pathogenic effects of aflatoxins is also reported.
I found the manuscript well prepared and of great interest, since the topic outlines pet health issues that deserve more attention from the consumers and the stakeholders. The global structure is well organized and clear to the reader.
Even if a strong English language revision is strictly required, I would suggest a style revision by a native speaker, in order to remove those small oddities that deteriorate the general good impression experienced while reading.
The bibliography is extremely updated and complete.
Author Response
Observation |
Location (Line No.) |
Attention to observations |
New Location (Line No.) |
Even if a strong English language revision is strictly required, I would suggest a style revision by a native speaker, in order to remove those small oddities that deteriorate the general good impression experienced while reading. |
|
The revised document was edited by native speakers Abby Greazel and Keith MacMillan (professional proofreading). |
|
Round 2
Reviewer 2 Report
I commend the authors for completing an extensive revision of the manuscript in a timely manner. All of my comments and suggestions were adequately addressed, and I recommend the manuscript for publication.